# RECREATE: a study protocol for a multicentre pilot cluster randomised controlled trial (cRCT) in UK stroke services evaluating an intervention to reduce sedentary behaviour in stroke survivors (Get Set Go) with embedded process and economic evaluations

Jennifer Airlie [ID],[1] Louisa-Jane Burton [ID],[1] Bethan Copsey [ID],[2] Coralie English [ID],[3,4] Amanda Farrin [ID],[2] Claire F Fitzsimons [ID],[5] Ivana Holloway [ID],[6] Judith Horrocks,[2] Jessica Faye Johansson [ID],[1,7] Gillian Mead [ID],[8] Lauren A Moreau [ID],[2] Seline Ozer [ID],[1] Anita Patel [ID],[9] Nahel Yaziji [ID],[10] Anne Forster [ID],[1,7] on behalf of the RECREATE Programme Management Group

For numbered affiliations see end of article.

**Correspondence to**
Dr Jennifer Airlie;
jennifer.airlie@bthft.nhs.uk

## ABSTRACT

**Introduction** Sedentary behaviour (sitting or lying during waking hours without being otherwise active) is strongly associated with adverse health outcomes, including all-cause, cancer and cardiovascular mortality in adults. Stroke survivors are consistently reported as being more sedentary than healthy age-matched controls, spending more hours sedentary daily and sustaining longer unbroken bouts of sedentary time. An evidence-based and clinically feasible intervention ('Get Set Go') was developed. A pragmatic definitive trial to evaluate Get Set Go was planned; however, due to the unprecedented effects of the COVID-19 pandemic on National Health Service (NHS) services this study was reduced in size and scope to become an external pilot trial. We report the protocol for this external pilot trial, which aims to undertake a preliminary exploration of whether Get Set Go is likely to improve ability to complete extended activities of daily living in the first year post-stroke and inform future trial designs in stroke rehabilitation.

**Methods and analysis** This study is a pragmatic, multicentre, two-arm, external pilot cluster randomised controlled trial with embedded process and economic evaluations. UK-based stroke services will be randomised 1:1 to the intervention (usual care plus Get Set Go) or control (usual care) arm. Fifteen stroke services will recruit 300–400 stroke inpatient and carer participants, with follow-up at 6, 12 and 24 months. The proposed primary endpoint is stroke survivor self-reported Nottingham Extended Activities of Daily Living scale at 12 months. Endpoint analyses will be exploratory and provide preliminary estimates of intervention effect. The process evaluation will provide valuable information on intervention fidelity, acceptability and how it can be optimised.

## STRENGTHS AND LIMITATIONS OF THIS STUDY

⇒ An external pilot cluster randomised control trial design enables the collection of feasibility and acceptability data to inform future trial designs in stroke rehabilitation.

⇒ The comprehensive set of outcome measures collected will ensure any potential impacts of the intervention on meaningful outcomes for stroke survivors (eg, short-term and long-term changes to activities of daily living, quality of life and well-being) are captured.

⇒ The study will collect objective measures of sedentary behaviour (using activity monitors), which to date have been infrequently assessed in post-stroke rehabilitation trials with longer-term outcomes up to 24 months.

⇒ The inclusion of an embedded process evaluation (reported separately) will provide valuable information on the intervention fidelity, acceptability and how it can be optimised.

⇒ The inclusion of a nested Study Within A Trial (reported separately) will enable the systematic evaluation of whether a video animation could enhance participant understanding and subsequently uptake, engagement and compliance with the intervention and thus inform how best to implement the intervention in practice.

**Ethics and dissemination** The study has been approved by Yorkshire and The Humber – Bradford-Leeds Research Ethics Committee (Ref: 19/YH/0403). Results will be disseminated through journal publications and conference presentations.

**Trial registration number** This trial was registered prospectively on 01 April 2020 (ISRCTN ref: ISRCTN82280581).

## INTRODUCTION
### Background and rationale

There are clear associations between sedentary behaviour in the general population and increased risk of adverse health outcomes, including all-cause, cancer and cardiovascular mortality, and incidence of cardiovascular disease, cancer and type 2 diabetes.[1 2] Sedentary behaviour is defined as any waking behaviour characterised by low energy expenditure ≤1.5 metabolic equivalent of task while in a sitting, lying or reclining posture,[3] and importantly differs from physical inactivity, which refers to achieving insufficient levels of moderate-to-vigorous physical activity (MVPA). An individual may therefore not reach the recommended levels of physical activity (PA) yet spend little time sitting, whereas others may be physically active for short bursts (eg, running for an hour), but spend prolonged periods sitting. Increasing MVPA can offset some of the detrimental effects of sedentary behaviour, although high levels are required (ie, >300 min/week).[1]

Stroke survivors are more sedentary than other population groups.[4–6] Longitudinal studies;[7] systematic reviews[8 9] and observational studies[6] reported that stroke survivors are considerably more sedentary (in hours per day) and have longer unbroken bouts of sedentary time than healthy age-matched controls. Sedentary behaviour does not improve over the first year after stroke[7] and appears independent of the level of functional recovery,[7 8 10] with physical ability only having a small influence on time spent sitting in people living at home 6 months after stroke.[10] The high amount of reported sitting time by stroke survivors is likely to generate a range of negative outcomes. Epidemiological studies[11] place them in the highest quartile for cardiovascular risk. For a stroke survivor, increased sedentary behaviour may thus result in a 'perfect storm' of deteriorating physical function and health-related quality of life, coincident with rising risks for cardiovascular disease.

There has been increasing focus in international public health guidance on reducing sedentary behaviour, particularly for those with disabilities, such as those resulting from stroke, as for many it may be a more achievable target than increasing PA.[12] Short brief activity breaks throughout the day may be as effective as a continuous 30-min or 60-min bout of exercise (summarised in a study by Dempsey *et al*[13]). Experimental studies (primarily short-term, laboratory-based work) provide supporting evidence of the positive effect on metabolic outcomes of breaking up sitting time with short bursts of activity in standing[14 15] and reducing stroke risk.[16]

Furthermore, the potential benefits of reducing sedentary behaviour reach beyond those relating to mortality and cardiovascular risk. Indeed, sedentary behaviour has been identified as a modifiable behaviour to enhance physical function[17] and maintain muscle strength, thus potentially supporting the ability to undertake activities of daily living (ADLs).[17–19] These benefits are important not just for stroke survivors themselves but may also extend to their carers; a recent systematic review identified post-stroke disability as the biggest predictor of carer burden.[20] Improved physical function in stroke survivors may thus reduce carer strain. Sedentary behaviour has also been identified as important in relation to mental health, cognitive outcomes and health-related quality of life.[21]

To address the challenge of reducing sedentary behaviour after stroke, we embarked on a programme of work (RECREATE) to develop an evidence-based and clinically feasible intervention to reduce/break up sedentary behaviour in stroke survivors. Intervention development was informed by systematic reviews and qualitative work, and underpinned by behaviour change theory (the Behaviour Change Wheel[22]). The final intervention ('Get Set Go') was developed using co-production methods.[23] A logic model of intended intervention mechanisms and outcomes was also constructed, and implementation strategies were refined using a case study approach. The intervention is intended to begin early after stroke (in the inpatient setting) and continue into the community, and is designed to fit into existing rehabilitation and recovery pathways.

This protocol represents the final stage in the RECREATE programme, which aims to evaluate the developed intervention in a sample of stroke survivors and carers. A large definitive multicentre cluster randomised controlled trial (cRCT) with 34 stroke services each recruiting 34 participants was initially planned, with the aim of evaluating the effectiveness and cost-effectiveness of the developed intervention. However, due to issues related to the worldwide COVID-19 pandemic, site recruitment was slower than expected. As a result, in spring 2022, following meetings with the Programme Steering Committee (PSC) and a stakeholder meeting convened by the funder (National Institute for Health and care Research (NIHR)), a decision was taken with the funder to amend the protocol to reduce the trial in size and scope to become an external pilot trial.

### Objectives

The primary objective is to undertake a preliminary exploration of whether the intervention (Get Set Go) is likely to improve the ability to complete extended activities of daily living (EADLs) in the first year after stroke. A key secondary objective is to explore whether the intervention is likely to reduce sedentary behaviour. Other secondary objectives are to:
1. Explore core resource use and costs associated with delivering the intervention including impacts on wider care.
2. Explore whether the intervention reduces cardiovascular risk factors.
3. Explore whether the intervention reduces disability.
4. Explore whether the intervention improves health and well-being outcomes, such as health-related quality of life and mental well-being.
5. Explore whether the intervention reduces carer strain.

**Table 1** Stroke service eligibility criteria

| Inclusion criteria | Exclusion criteria |
|---|---|
| Both the inpatient and community service agree to participate. | Previous participation in research leading to the development of the intervention. |
| Availability of research staff to undertake participant recruitment. | Currently implementing/intending to implement similar interventions within trial duration. |
| Agreement that recruitment targets are feasible/acceptable (considering patient throughput). | Early supported discharge (ESD) (or community if no ESD) waiting list >4 weeks post-discharge. |
| The intervention can be feasibly implemented (if so randomised). | |

6. Explore and understand implementation of the intervention, including intervention adherence, compliance, the process, benefits and challenges.
7. Explore and understand how the intervention is experienced and understood by recipients and providers.
8. Explore potential moderators and mediators of the intervention effect.

Objectives 1–5 will be addressed in the cRCT. Additional objectives addressed in the cRCT relate to gathering feasibility and acceptability data to inform future trial designs in stroke rehabilitation. Objectives 6–8 will be addressed in an embedded mixed-methods process evaluation (protocol reported separately), which will develop an understanding of how the intervention is implemented, understood and experienced by both providers and recipients.

## METHODS
### Design
This trial is a pragmatic, multicentre, two-arm external pilot cRCT. The Get Set Go intervention will be delivered alongside usual care at the level of the stroke service (ie, cluster). The cluster design will reduce between-group contamination. Eligible stroke services will be randomised on a 1:1 basis to the two arms of the trial:

either intervention (usual care plus Get Set Go) or control (ie, usual care). All stroke survivors in services allocated to the intervention arm will receive Get Set Go; however outcomes will only be measured for those who consent to trial participation.

### Settings
The trial will be conducted in up to 15 National Health Service (NHS) stroke services across the UK. A stroke service is defined as an acute and/or rehabilitation stroke unit with a linked community service over a defined geographical area. Service eligibility criteria are detailed in table 1.

### Recruitment
Three to four hundred stroke survivors and carers will be recruited to participate. Eligibility criteria are outlined in table 2. Carer involvement is not a requirement for stroke survivor inclusion.

Conversations with interested stroke services began in October 2019 in preparation for the study starting in December 2019. Recruitment of stroke survivors and carers started in January 2021. The study is projected to complete recruitment by the end of April 2023 with the follow-up due to complete in April 2024. The end of the

**Table 2** Stroke survivor and carer eligibility criteria

| | Inclusion criteria | Exclusion criteria |
|---|---|---|
| Stroke survivor | Aged ≥16 years at time of stroke. | Receiving palliative care. |
| | Clinical diagnosis of new or recurrent ischaemic or haemorrhagic (excluding subarachnoid haemorrhage) stroke. | Due to be discharged outside the defined geographical area of the associated community service(s) participating in the trial. |
| | Requires manual contact of no more than one person to stand to prevent falling (manual contact consists of continuous or intermittent light touch to assist balance or coordination, that is, not to support body weight). | |
| | Plan to live in the community post-discharge. | |
| | Informed consent/consultee declaration is provided. | |
| Carer | Aged ≥16 years. | Stroke survivor does not consent to participate. |
| | Family member or friend regularly engaging with a stroke survivor participant (≥once per fortnight). | |
| | Provide informed consent. | |

study is defined as the date the last participant's data item is collected.

## Interventions

All stroke survivor participants (irrespective of randomisation allocation) will receive usual care within an organised stroke service. In line with the pragmatic nature of the trial, the comparator is 'usual care' as determined by local policy and practices, to draw comparison with current practice in NHS stroke services, especially given the variation in pathways. Information on the care delivered by both control and intervention sites will be captured by survey and enhanced by conversations with clinical staff (if further details are required) prior to randomisation and at regular intervals (approximately every 3–6months) until the end of the trial. Observations conducted as part of the embedded process evaluation will provide additional insights into the nature of usual care provision at both control and intervention sites. This information will also provide context for trial implementation and monitor any potential contamination and/or confounding between the two arms of the trial.

In stroke services allocated to the intervention arm, participants will also receive the Get Set Go intervention. Get Set Go is a whole-service intervention designed to be implemented and embedded within routine practice. Delivery commences in the inpatient stroke unit setting and continues into the community for at least 12 weeks post-discharge. The intervention focuses on:

1. Educating staff and stroke survivors (and their family/friends/carers where appropriate) about the importance of standing and moving after stroke.
2. Preparing and enabling staff to support and encourage stroke survivors to stand and move more in everyday stroke care (as part of routine practice).
3. Encouraging stroke survivors to monitor their own standing and moving, with assistance from family/friends/carers where appropriate.

As Get Set Go will be delivered at service level, all clinical staff members working within the stroke services randomised to deliver the intervention (ie, inpatient and community settings) will be invited to attend a training session (~1 hour) to prepare for delivering Get Set Go. This will include the intervention rationale and an overview of key components. Staff will participate in practical tasks to ensure they feel confident in supporting and encouraging stroke survivors, who are capable of standing independently/with the assistance of one person, to stand and move more in everyday stroke care (ie, as part of routine clinical practice). A Template for Intervention Description and Replication checklist[24] will be published with trial findings. There are no special criteria for discontinuing or modifying allocated interventions. Recommendations for standing and moving are made by staff as part of delivering the intervention alongside usual care in the stroke unit or community setting, based on their usual assessment techniques and clinical judgement. These recommendations will be regularly reviewed and modified in line with stroke survivors' individual capabilities and circumstances.

There are no provisions in place for post-trial care beyond the standard care provided within the participating NHS stroke services.

## Study Within A Trial

This cRCT will also incorporate a Study Within A Trial (SWAT) (protocol reported separately),[25] to evaluate whether a video animation could increase participant understanding of the intervention, and result in improved engagement and compliance, potentially resulting in improved outcomes. Intervention sites will be randomised 1:1 to the SWAT intervention (Get Set Go plus video animation) or control (Get Set Go with no video animation). Results will be reported separately.

## Strategies to improve intervention adherence

The implementation team will maintain regular contact with clinical leads in sites allocated to the intervention arm to monitor delivery of the intervention and provide support as required. Clinical staff who are responsible for delivering the intervention will also complete monitoring records for all stroke survivors (whether or not they are trial participants) to document whether key tasks relating to the intervention have been completed (no personal data will be recorded for non-trial participants). It will be advised that a nominated member of staff checks these records have been completed on a regular basis (eg, towards the end of each shift). Regular review of these records will enable the study team to monitor intervention adherence and provide insight into intervention delivery. A short postal questionnaire completed by stroke survivor trial participants at 12 weeks post-hospital discharge will also assess intervention compliance, with those in the intervention arm responding to questions about the intervention; all participants will be asked about falls and use of a walking aid. Data on compliance will be analysed using descriptive summary statistics by allocation. Finally, intervention fidelity will be explored separately within the process evaluation, through site-based observations of intervention delivery, interviews with those delivering and receiving the intervention and documentary analysis (including completed intervention materials).

## Endpoints

The primary endpoint is stroke survivor self-reported EADLs at 12 months post-registration. The key secondary endpoint is mean daily sedentary time (minutes) at 12 months post-registration using activity monitor (activPAL) data, which provides a valid measure of posture and transitions in people with impaired mobility.[26] Other secondary endpoints at 6, 12 and 24 months post-registration (unless stated otherwise) for stroke survivor participants are listed below:

► Sedentary behaviour measured by activity monitor.

- EADLs (at 6 and 24 months only) measured by Nottingham Extended Activities of Daily Living scale (NEADL).[27]
- Health and disability measured by WHO Disability Assessment Schedule 2.0 (12-item).[28–32]
- Mental well-being measured by Warwick-Edinburgh Mental Well-being Scale.[33]
- Health status measured by European Quality of Life 5-Dimension Health Questionnaire (5 levels; EQ-5D-5L).[34]
- Quality-adjusted life year (QALY) gains (as above).
- Fatigue as measured by Fatigue Assessment Scale.[35 36]
- Death.
- Falls and use of a walking aid (12 weeks post-discharge, and at 6, 12 and 24 months).
- Self-reported sedentary behaviour measured by the Measure of Older Adults' Sedentary Time[37] and a Sedentary Behaviour Visual Analogue Scale (adapted from Chastin *et al*, 2019[38]).
- Institutionalisation, hospital readmission and emergency department attendance rates.
- Cardiovascular risk markers: Body mass index (BMI), waist circumference, blood pressure (BP).
- Total major vascular events: Composite measure of non-fatal stroke, non-fatal myocardial infarction or death due to any vascular cause (including unexplained sudden death).[39]
- Self-reported health and social care service use and informal care inputs measured by a Client Services Receipt Inventory (CSRI) specifically adapted for this trial from versions used in previous stroke rehabilitation trials.[40 41]
- Costs from a health and social care perspective, and a societal perspective that further includes informal care.

The carer endpoint is caregiver burden measured by Modified Caregiver Strain Index[42] at 6, 12 and 24 months.

An overview of which assessments are undertaken at each time point and the method of completion is provided in table 3. Information about the validity and reliability of the measures used is available in online supplemental appendix 1. Stroke survivors will also complete an additional brief postal questionnaire at 12 weeks post-discharge.

### Randomisation and blinding

Site randomisation will be performed centrally by the Clinical Trials Research Unit (CTRU) and the allocation sequence will be computer-generated using a minimisation programme incorporating a random element to ensure the treatment arms are well-balanced. Eligible stroke services will be randomised on a 1:1 basis either to intervention or control, stratified by:

1. Sentinel Stroke National Audit Programme (SSNAP) grading level:[43] A versus B/C/D/E (based on the latest available audit data at the time of randomisation: A=first class service, B=good/excellent in many aspects, C=reasonable overall, some areas require

improvement, D=several areas require improvement and E=substantial improvement required).
2. Stroke service composition: Single or combination of trusts.
3. Stroke unit size (acute): Median cut-off of ≤24 and >24 beds based on SSNAP acute organisational audit data (type 2 (solely for patients >72 hours post-stroke) and type 3 beds (used for both pre-72-hour and post-72-hour care) combined—based on the SSNAP audit data and confirmation by site).

Concealment of sequencing is ensured by the physically separate locations of the CTRU (where the randomisation outcome is stored) and the blinded study researchers. Following the randomisation, unblinded members of the study team, the principal investigator at the site and clinical staff who will be involved in intervention delivery, will be informed of the randomisation outcome via secure email to facilitate organisation of intervention training in services allocated to this arm. Local researchers responsible for participant recruitment will be notified that randomisation has taken place but will remain blinded to randomisation outcome and will operate independently of the clinical staff involved in intervention delivery.

Local researchers involved in participant recruitment and in-person data collection (including at baseline, administering activity monitors and assessment of cardiovascular risk markers) will be blinded to the treatment allocation for their site. At the follow-up time points, participant-completed outcome measures will be administered by CTRU staff via postal questionnaire, eliminating the potential for interviewer bias. The activity monitor (on which data is recorded and stored until download) is sealed with no external indication of recording therefore it is not open to bias. Finally, data on mortality, readmissions and use of institutional care will be collected from independent sources and are unlikely to be subject to bias. Instances of unblinding will be recorded via staff self-report.

Participants, clinical staff involved in intervention delivery and the statistical team analysing the outcome data, will not be blinded to treatment allocation. However, to minimise treatment bias, clinical staff involved in intervention delivery will not be overtly informed of which patients are participating in the trial. Additionally, to monitor for potential selection bias, key characteristics of included trial participants will be compared with publicly available data sets (eg, SSNAP[43]) to monitor for evidence of selection bias.

### Procedure

Eligible stroke survivors will be initially approached while they are an inpatient by a local researcher, who will provide verbal and written information about the trial before seeking informed consent. Examples of an information sheet and informed consent form for stroke survivor participants are provided in online supplemental appendices 2–4.

**Table 3** Overview of assessments and method of completion at each study time point

| Outcome/assessment | Method of completion | Time point | | | |
|---|---|---|---|---|---|
| | | Baseline | 6 months post-registration | 12 months post-registration | 24 months post-registration |
| Stroke survivor | | | | | |
| Demographic details | Self-report | X | | | |
| Nottingham Extended Activities of Daily Living Scale[27] | Self-report | X | X | X | X |
| WHO Disability Assessment Scale 2.0 12-item[28–32] | Self-report | X | X | X | X |
| Warwick-Edinburgh Mental Well-being Scale[33] | Self-report | X | X | X | X |
| European Quality of Life 5-Dimension Health Questionnaire[34] | Self-report | X | X | X | X |
| Fatigue Assessment Scale[35 36] | Self-report | X | X | X | X |
| Measure of Older Adults' Sedentary Time[37] | Self-report | X | X | X | X |
| Sedentary Behaviour Visual Analogue Scale (adapted from 38) | Self-report | X | X | X | X |
| Use of a walking aid | Self-report | X | X | X | X |
| Falls question | Self-report | X | X | X | X |
| Hospital admissions | Self-report/ electronic health records | | X | X | X |
| Client Service Receipt Inventories to collect data on health and social care use and informal care inputs (adapted from 40 41) | Self-report | X | X | X | X |
| Total major vascular events[39] | Electronic health records via routine data providers | X | X | X | X |
| Sedentary behaviour—activPAL measurements | Researcher administration/ post | X | X | X | X |
| Risk markers for cardiovascular disease (height* and weight for body mass index; waist circumference; blood pressure) *Collected at baseline only | Researcher collection from notes/ researcher visit/ self-completion | X | X | X | X |
| Adverse events | Self-report/ electronic health records | | X | X | X |
| Survival status | Researcher collection from electronic health records | | X | X | X |
| Admission to institution | Self-report | | X | X | X |
| Carer | | | | | |
| Demographic details | Self-report | X | (X) | (X) | (X) |

Continued

**Table 3** Continued

| Outcome/assessment | Method of completion | Time point | | | |
|---|---|---|---|---|---|
| | | Baseline | 6 months post-registration | 12 months post-registration | 24 months post-registration |
| Modified Caregiver Strain Index[42] | Self-report | | X | X | X |
| Employment/occupation status | Self-report | X | X | X | X |

(X) only collected if 'new carer'.

To ensure that the trial population is representative of the clinical stroke population and inclusive of stroke survivors with cognitive impairment (including comprehension or language difficulties), recruitment procedures will include the use of accessible materials and procedures for consultee declaration (where appropriate), in compliance with the Mental Capacity Act 2005 (MCA).[44] As per the MCA, each stroke survivor will be assumed to have capacity unless it is established that they lack capacity. If there is any concern about capacity the local researcher will consult ward staff and/or a family member/carer (as appropriate) and a collective decision will be made as to whether the stroke survivor is deemed to have capacity to consent to participation in the trial. Changes in capacity will be monitored throughout the trial.

Potential stroke survivor participants who are deemed to have capacity to consent will be provided with written and verbal information about what their participation in the study would involve. Following time to consider participation (including discussion with family and/or health professionals), those wishing to take part will provide informed consent through signing (or making a mark), or if they are unable, through witnessed verbal consent (by an independent observer). Should a potential participant be discharged prior to consent being gained and they consent to being contacted by the researcher, consent procedures will take place in their own home.

Should stroke survivors be deemed to lack capacity to provide informed consent, the researcher will seek to identify 'a supporting family member, friend or carer' (defined as someone who is in regular (at least weekly) contact with the stroke survivor), to support intervention delivery and outcome assessment. This individual will not necessarily be enrolled in the study themselves, unless they choose to enrol as a carer participant. If such a supporter is identified, the local researcher will either: act in accordance with an advance directive relative to research participation (if available) or seek advice from a personal consultee (through consultee declaration). If a personal consultee is approached, they will be provided with an additional information sheet outlining their role. Consultees will provide assent in person or via telephone (and signed by the local researcher, eg, if the consultee is unable to visit the hospital prior to the stroke survivor's discharge). Carer participants will be approached by the local researcher once stroke survivors with capacity have

given their verbal agreement or following provision of a consultee declaration for a stroke survivor participant who lacks capacity. Those wishing to participate will be asked to provide informed consent using the same procedure described above. Carers lacking the capacity to provide written informed consent will be excluded.

The local researcher will administer baseline questionnaires to stroke survivor and carer participants face-to-face and offer support where required. Proxy completion of questionnaires by a carer, family member or friend will be permitted (for those with and without capacity), with separate proxy questionnaires provided if required. Stroke survivors will also be asked to continuously wear an activPAL activity monitor on the thigh of their strongest/unaffected leg for 9 consecutive days. The monitor will be attached by the local researcher. They will also be asked to complete a purposely developed daily diary. This will capture if/when the monitor has been removed; the time they went to bed and got up each day and whether they had episodes of sleep (ie, naps) during the day. This information will assist with analysis and facilitate monitoring of adverse reactions to the materials used to attach the monitor. Local researchers will also collect data on risk markers for cardiovascular disease through face-to-face assessment or retrieval from medical records. These include height and weight (for calculation of BMI), waist circumference and BP.

At the follow-up time points (6, 12 and 24 months post-registration), participant-completed outcome assessments will be administered by post in an assessment pack, including a covering letter and a prepaid envelope for postal return. Activity monitors may be administered face-to-face by a local researcher or alternatively, be sent by post to the participant and returned via prepaid envelope. Cardiovascular risk markers (with the exception of height) will be collected alongside the activity monitor assessment. If it is not possible to organise a face-to-face visit, with the participants' agreement, options for self-assessment will be explored. Guidance will be provided to local researchers as to the action required if a BP reading is abnormal, including reporting this to a named clinician or notifying the participant's general practitioner (GP). The participant will also be encouraged to discuss this with their GP.

At each participant follow-up, changes in circumstances, including change of address, institutionalisation

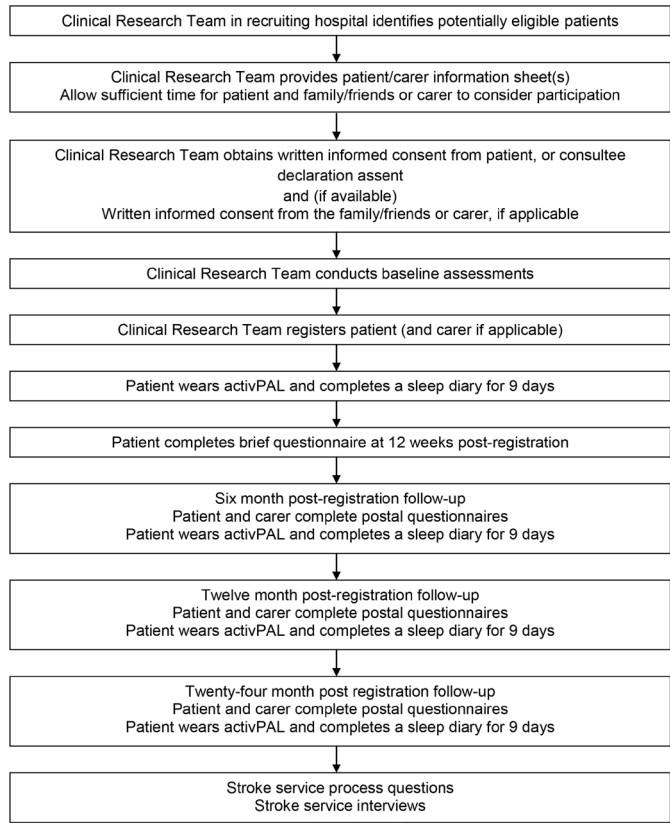

**Figure 1** Study timeline for participants.

and death will be obtained from either recruiting teams, healthcare providers, participants or their relatives. Additionally, for stroke survivors, the following data will be collected from electronic health records via routine data providers or directly from the recruiting hospital at each time point: total major vascular events (composite of non-fatal stroke; non-fatal myocardial infarction; or death due to any vascular cause (including unexplained sudden death));[39] death; hospital admissions; emergency department attendance and institutionalisation. Stroke survivors will be made aware of the use of routine data providers (including NHS England, SystmOne, EMIS Web and other relevant providers) at the time of study entry.

The study timeline for participants is presented in figure 1.

Initial letters, reminder letters and a participant pathway diagram will be distributed to participants to maximise data return at all time points. The questionnaire sent to assess intervention compliance at 12 weeks post-hospital discharge will also serve as a reminder of study participation and act as a retention tool. At the follow-up time points, reminders will be administered via telephone, text message, email or post (according to participant preference) should the questionnaire not be returned within 2 weeks. If outcome assessments (with particular emphasis on the NEADL as the primary outcome), cannot be obtained by post, then (where appropriate) the assessments may be administered over the telephone or face-to-face to maximise return of data. Missing data, except

individual data items collected via the postal assessment packs, will be sought until: it is received; confirmed as not available; or the trial is at analysis.

In instances where a participant withdraws from the trial clarification will be sought as to whether this is from all trial activities or individual components, for example, wearing the activity monitor or consenting to be contacted about participation in process evaluation activities. Unless participants explicitly instruct otherwise, electronic health record data will continue to be collected.

## Data management
Data will be monitored for quality and completeness using established verification, validation and checking processes, for example, double data entry, and securely stored. Received activPAL data will be subject to quality checks by researchers and where needed, additional details to aid interpretation will be sought from the recruiting site and/or participant diaries. All data collected during the course of the trial will be kept strictly confidential using established processes. Data will be securely archived for 10 years after the completion of the study and then destroyed.

## Statistical analysis
All analyses and data summaries will be conducted on the intention-to-treat population (ITT),[45] defined as all participants registered to the trial, regardless of non-compliance with the protocol or withdrawal from the study or losses to follow-up. Data from all participants recruited within a stroke service will be analysed according to the randomised allocation for that stroke service. No formal interim or subgroup analyses are planned, and final analysis will take place when all available data have been received.

### Sample size
As previously described, our original plan for a large definitive multicentre RCT was reduced in size and scope, due to issues related to the COVID-19 pandemic. We aim to recruit 300–400 participants across 15 stroke services. This revised sample size is based on the anticipated recruitment rate within the remaining time available in the funded programme, estimated at two participants per month per site. As intervention effectiveness will not be evaluated and all analyses will be of an exploratory nature, the study does not have a formal power calculation.

### Primary endpoint analysis
In line with study objectives, all endpoint analyses will be exploratory and provide preliminary estimates of intervention effect. The primary outcome will be compared between randomised groups using multilevel mixed effects model[46] adjusting for design effects and other relevant known predictors of outcome, with participants nested within services, with services treated as random effects. If appropriate, missing data will be multiply imputed at the individual participant level. Parameter estimates will be reported with 95% CIs. Sensitivity analyses of the

primary endpoint will be conducted to assess the impact of missing data, the choice of imputation model and of assuming data are missing not at random, as appropriate. For the primary analysis, missing data will be assumed missing at random.

## Incorporating the SWAT into the statistical analysis

For the primary analysis, the SWAT intervention will be accounted for in the regression model using an interaction term. Sensitivity analyses will explore the impact of the SWAT on the Get Set Go intervention effect, for example, analysing the outcomes as a three-arm study separating the two SWAT arms. This will be described in further detail in the statistical analysis plan.

## Secondary endpoints analyses

Secondary outcomes will be analysed in a similar manner to the primary outcome. For secondary outcomes, summary statistics will be presented for each time point by treatment group (means, SD, medians, minimum, maximum and quartiles for continuous variables, and counts and percentages for categorical variables). Secondary outcomes will be analysed using the same approach as the primary outcome with the relevant model for the type of outcome variable using multilevel linear or logistic regression, with multiple imputation for missing data where appropriate. Continuous distributions will be transformed where residuals are non-normal.

Self-reported falls, institutionalisation, readmission, cardiovascular risk markers, total major vascular events and death rates between treatment arms will be summarised descriptively.

Quantitative summaries of intervention delivery will evaluate uptake of the intervention, adherence to the processes, staff and participant engagement and quality of intervention delivery, overall and by each site. Data on usual care will be summarised descriptively.

## Activity monitor (activPAL) data

Sedentary time, as measured by activPAL activity monitors, will be analysed as total sedentary minutes on an average day as well as the percentage of sedentary time during wake time. Data from the monitors will be automatically recorded and processed using the activPAL software (PAL Technologies). We will process data, remove sleep time and calculate, for each day, the total sedentary time; total sitting time accumulated in bouts ≥30 min, and bouts ≥60 min, number of sedentary bouts ≥30 min and bouts ≥60 min, total standing time, total stepping time and number of sit-to-stand transitions.

We will then calculate the key secondary outcome of mean daily sedentary time for the days when the monitor was worn. All other sedentary behaviour secondary outcomes will be calculated in the same way. The sedentary behaviour outcomes would ideally be based on 7 days of continuous wear of the monitor. The calculation of sleep time, definition of a valid wear day and number of valid wear days required for inclusion in the analysis will be informed by the results of earlier feasibility work.

## Further secondary analyses

Potential moderator variables: Potential predictors of response to the intervention will be explored via inclusion of key baseline stroke survivor characteristics (identified in our earlier work) as interaction effects in the primary analysis model, to understand who might benefit from the intervention.

Potential mediator variables will be explored by modelling the relationship between process variables (eg, intervention adherence, number/frequency of prompts and visits from staff, participant self-reported use of the guide, mediators such as fatigue or mood) and the primary outcome.

## Methods in analysis to handle protocol non-adherence and any statistical methods to handle missing data

All analyses and data summaries will be conducted on the ITT population. Missing data is anticipated at the item, scale and assessment levels. Missing data on key variables/questionnaires will be explored. Missing data will be multiply imputed, where appropriate. Reasons for missing participant assessments will be explored and considered in the imputation process (eg, died, moved away). If someone dies, they will be classed as lost to follow-up. When individual scales within the assessment questionnaire have missing items, scoring instructions specific to the scale will be followed, or if no such instructions exist, the summary scores will be calculated from the non-missing items using multiple imputation under the missing at random mechanism, as long as at least 75% items are non-missing. We will test missing data assumptions in a sensitivity analysis.

## Health economic analysis

The embedded health economic evaluation will assess resource use, costs and outcomes to inform data collection approaches for a future definitive trial, specifically:

► Alternative approaches to measuring hospital resource use (routine records vs self-report);
► A short (6-month) versus medium (12-month) versus longer (24-month) analytical time horizon; and
► A reduced list (narrow perspective) evaluation based on routine data versus long list (comprehensive perspective) evaluation incorporating self-reported data.

Two perspectives will be considered, as relevant to the analyses: health/social care (including institutionalisation), and a wider perspective covering health/social care (including institutionalisation) plus informal care given the significance of such inputs for this patient group.

Resource use data will be collected using a combination of NHS England secondary care data and self-report approaches via the CSRI. Resource use will be combined with relevant unit costs, using within-programme estimates related to the intervention, national estimates for

other health and social care resources and both an opportunity cost and replacement cost approach in turn for informal care. Total costs will be computed for each individual according to each of the estimation alternatives set out above. Differences between the two trial arms will be compared using bootstrap regression methods due to the expected skewness in data distributions.

Exploratory cost-effectiveness analyses will combine total costs with the primary outcome and with QALY gains estimated by attaching relevant general population utility weights to EQ-5D-5L health states at each time point. Uncertainty will be analysed using cost-effectiveness planes and cost-effectiveness acceptability curves. Sensitivity analyses will explore the potential impact on cost-effectiveness from different implementation/costing scenarios. This would better inform future trial design and implementation discussions. Other relevant sensitivity analyses will be determined during the study and will be specified in the analysis plan prior to analyses.

The economic analyses will follow the same broad principles as the outcomes analyses, for example, use of a signed off detailed analysis plan prior to analysis, adherence to ITT principles, imputation to account for missing values as appropriate, inclusion of baseline and other covariates for the comparisons, accounting for the clustering of patients within stroke services.

## Oversight and monitoring

This trial is the final study in a 7-year Programme Grant for Applied Research funded by the NIHR. As such, a Programme Management Group (PMG), chaired by the chief investigator (CI) and attended by the programme co-applicants meets at least every 4 months, for strategic oversight and management of the programme, and to monitor progress against key objectives. An independent PSC chaired and attended by independent experts meets as a minimum every 12 months, and will provide external guidance and monitor progress. Finally, a Trial Management Group comprised of the CI, key co-applicants, study researchers and key members of the project delivery team at the CTRU will be convened to meet every month to oversee day-to-day management of the trial. A Data Monitoring Committee is not required due to the lack of serious adverse events (SAEs) anticipated to be due to the intervention. The PSC will take on the role of monitoring safety concerns.

## Adverse event reporting and harms

Adverse events (AEs) such as falls and musculoskeletal injuries represent an inherent consequence of an active rehabilitation process and therefore cannot be entirely avoided. Similarly, in this patient population, acute illness resulting in hospitalisation, new medical problems and deterioration of existing medical problems are expected. In recognition of this, events fulfilling the definition of an AE or SAE will not be reportable in this study unless they result from administration of any research procedures and therefore fulfil the definition of an unexpected and

related SAE, or if they constitute one of the following: death, hospital readmission, institutionalisation, treatment on an emergency outpatient basis or emergency department attendance. The latter will be reported at 6, 12 and 24 months post-registration follow-up by the clinical research team using health and social care records. All related/unexpected SAEs occurring from the date of consent up to 24 months post-registration must be reported within 24 hours of the clinical research staff becoming aware of the event. All related/unexpected SAEs will be reviewed by the CI and subject to expedited reporting to the sponsor and the main Research Ethics Committee (REC) by the CTRU on behalf of the CI within 15 days.

## Patient and public involvement

Patient and public involvement (PPI) is central to this research. The research question was informed by initial discussion with our stroke survivor and carer group, who highlighted the need for interventions to encourage active movement after stroke and contributed to grant development. PPI is subsequently being sought at every stage of the programme of research. Specific input from our PPI representatives includes: being actively involved in the design and management (eg, steering/advisory group) of this research; assisting with developing participant information resources (including trial documents and intervention materials); contributing to ongoing reporting of the study and dissemination of emerging research findings.

## ETHICS AND DISSEMINATION

The current protocol (V.9.0, 10 August 2022) received ethical approval from the Yorkshire and The Humber – Bradford-Leeds REC (Ref: 19/YH/0403). Protocol amendments are processed in line with Health Research Authority and REC guidelines and processes. The results of the study will be published in peer-reviewed publications and will be presented at relevant national and international conferences. Authorship will be agreed in accordance with International Committee of Medical Journal Editors recommendations and the publication policy that has been agreed by the PMG for this programme of research. In collaboration with patient and public involvement representatives, lay reports to disseminate research findings to patient groups and clinical staff at participating sites will also be compiled.

**Author affiliations**
[1]Academic Unit for Ageing and Stroke Research, Bradford Institute for Health Research, Bradford Teaching Hospitals NHS Foundation Trust, Bradford, UK
[2]Leeds Institute of Clinical Trials Research, Clinical Trials Research Unit (CTRU), University of Leeds, Leeds, UK
[3]School of Health Sciences, The University of Newcastle, Newcastle, New South Wales, Australia
[4]Heart and Stroke Research Program, The University of Newcastle Hunter Medical Research Institute, Newcastle, New South Wales, Australia
[5]Physical Activity for Health and Research Centre, Institute for Sport Physical Education and Health Sciences, University of Edinburgh, Edinburgh, UK

[6]Institute of Medical Psychology and Medical Sociology, University Medical Center Göttingen, Waldweg, Göttingen, Germany
[7]Academic Unit for Ageing and Stroke Research, Leeds Institute of Health Sciences, University of Leeds, Leeds, UK
[8]Centre for Clinical Brain Sciences, University of Edinburgh, Edinburgh, UK
[9]Anita Patel Health Economics Consulting Ltd, London, UK
[10]Institute of Psychiatry, Psychology and Neuroscience, King's College London, London, UK

**Acknowledgements** The authors would like to acknowledge the support of the Clinical Research Network and also express our appreciation for all of the hard work and support provided by the clinical teams who are helping to deliver this trial. The authors would also like to thank members of the Programme Steering Committee for their helpful contributions through the RECREATE programme of research. Finally, the authors would like to acknowledge the expert support of the RECREATE Programme Group who meet regularly to oversee the programme of research.

**Collaborators** RECREATE Programme Management Group: Jennifer Airlie, Karen Birch, Gillian Carter, Bethan Copsey, Florence Day, Coralie English, Amanda Farrin, Alison Fergusson, Claire Fitzsimons, Anne Forster, Jessica Faye Johansson, Rebecca Lawton, Laura Marsden, Gillian Mead, Lauren Moreau, Seline Ozer, Anita Patel, Rosemary Shannon, Nahel Yaziji.

**Contributors** AFo is lead grant holder and chief investigator and will oversee the design and implementation of the trial. CE, CFF and GM are co-investigators who have contributed to the development of the protocol, and attended regular programme meetings, providing advice as needed. AFa, assisted by IH and BC, led the statistical and methodological design of the trial and is the statistical guarantor responsible for the main statistical analysis and reporting. AP leads the design of the health economic components of the trial and is responsible for the health economic analyses, assisted by NY. LAM, assisted by JH, leads on data management and supports the day-to-day operation of the trial. SO, assisted by JA, is responsible for managing implementation of the intervention and delivery of the trial. L-JB also assisted with managing delivery of the trial. JFJ leads the embedded process evaluation. JA leads on activPAL data implementation and data issues. JA and L-JB drafted the manuscript. All authors read and approved the final manuscript. This paper is written on behalf of the RECREATE Programme Management Group who meet regularly to oversee the programme of research. All authors reviewed and approved the final manuscript

**Funding** This work was supported by the National Institute for Health and Care Research (NIHR) under its Programme Grants for Applied Research Programme, grant number RP-PG-0615-20019. The views expressed are those of the author(s) and not necessarily those of the NIHR or the Department of Health and Social Care.

**Competing interests** AFo has received additional research grant support from NIHR through the following funding streams: Senior Investigator award, Health Technology Assessment and Health and Social Care Delivery Research (HS&DR). AFo has previously received support from the Stroke Association to attend the UK stroke forum and received payment from the National Institute for Health (USA) for panel membership (2021 and 2022). AFo is currently the chair/a member of the programme steering committees for NIHR research programmes (Grant reference numbers: NIHR 202339 and NIHR 202020) and has served on the following panels: NIHR Doctoral Fellowships, NIHR senior investigators committee (2019/2020), NIHR HS&DR committee (2016–2018) and Stroke Association Funding. CE has received grant funding from the Netherlands Organisation for Scientific Research (NOW) Taskforce for Applied Research (SIA RAAK) for work in a similar area (ie, sitting less and moving more after stroke) and is a non-executive Director representing interests of Research and Chair of Research Advisory Committee for the Stroke Foundation of Australia (unpaid). CFF is a co-investigator/collaborator on other grants on the topic of sedentary behaviour/physical activity and is therefore partially supported by grant funding received from the University of Edinburgh and the Irish Health Board. CE has previously been supported to conduct work in a similar area by grant funding received from the Chief Scientist Office of the Scottish Government, Medical Research Council Public Health Intervention Development award and the University of Edinburgh.

**Patient and public involvement** Patients and/or the public were involved in the design, or conduct, or reporting, or dissemination plans of this research. Refer to the Methods section for further details.

**Patient consent for publication** Not applicable.

**Provenance and peer review** Not commissioned; externally peer reviewed.

**ORCID iDs**
Jennifer Airlie http://orcid.org/0000-0002-7505-3049
Louisa-Jane Burton http://orcid.org/0000-0003-3617-1410
Bethan Copsey http://orcid.org/0000-0001-9783-6549
Coralie English http://orcid.org/0000-0001-5910-7927
Amanda Farrin http://orcid.org/0000-0002-2876-0584
Claire F Fitzsimons http://orcid.org/0000-0001-6192-1397
Ivana Holloway http://orcid.org/0000-0002-9542-883X
Jessica Faye Johansson http://orcid.org/0000-0003-3622-9598
Gillian Mead http://orcid.org/0000-0001-7494-2023
Lauren A Moreau http://orcid.org/0000-0002-0280-6345
Seline Ozer http://orcid.org/0000-0002-5791-5469
Anita Patel http://orcid.org/0000-0003-0769-1732
Nahel Yaziji http://orcid.org/0000-0002-2273-7957
Anne Forster http://orcid.org/0000-0001-7466-4414

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
