## [Reviewer comments · BMJ Open]

ARTICLE DETAILS

TITLE (PROVISIONAL)	RECREATE: A study protocol for a multi-centre pilot cluster Randomised Controlled Trial (cRCT) in UK stroke services evaluating an intervention to reduce sedentary behaviour in stroke survivors (Get Set Go) with embedded process and economic evaluations
AUTHORS	Airlie, Jennifer; Burton, Louisa-Jane; Copsey, Bethan; English, Coralie; Farrin, Amanda; Fitzsimons, Claire F.; Holloway, Ivana; Horrocks, Judith; Johansson, Jessica; Mead, Gillian; Moreau, Lauren; Ozer, Seline; Patel, Anita; Yaziji, Nahel; Forster, Anne

VERSION 1 – REVIEW

REVIEWER	Sánchez-Sánchez, M Luz University of Valencia
REVIEW RETURNED	13-May-2023

GENERAL COMMENTS	I would like to thank the authors for their interest in addressing a topic of such relevance and at the same time so complex. I would just like to make some suggestions to improve understanding and the possibility of replicating the study in other countries. First, the introduction deals with sedentary behavior and physical inactivity; however, the main objective includes extended activities of daily living. I believe that introducing why this variable is included as the main variable would be appropriate. I mean explaining in the introduction why it is relevant in a program that aims to reduce sedentary behaviors. Similarly, caregivers are not mentioned in the introduction section either, what would be the hypothesis in this regard? what is the reason for including them as participants in this study? I understand that the caregiver is a fundamental pillar to increase physical activity levels, but why evaluate their burden instead of their involvement/adherence to the program? Secondly, by conducting the study in different centers, have you considered the weight that may have the usual intervention differs greatly from one center to another, have you thought to control for this variable or at least record it? Thirdly, in relation to the sample size, the reason for not making a calculation should be explained: will the results of this pilot study be used to make a calculation of the subsequent sample size? Finally, in the manuscript the intervention is explained in a summarized manner, which makes it difficult to replicate the intervention protocol; it would be interesting to have an appendix with more information about the intervention protocol. In this regard, the subsection on the inclusion of a SWAT creates confusion: what does this part have to do with the general
---

	intervention? Will all the participants of the intervention group or only some randomly selected ones watch the animation video? If so, it is not explained how this circumstance will be integrated into the statistical analysis.
--	---

REVIEWER	Lin, Shuanglan Chongqing Medical University
REVIEW RETURNED	17-May-2023

GENERAL COMMENTS	This is a well-designed cluster RCT that aims to evaluate the Get Set Go intervention to reduce sedentary behaviour in stroke survivors. The authors provided a rationale research method. The details of the interventions, application process, outcomes assessments, and data analysis strategies were clarified well in the manuscript. However, some details about the participants' inclusion criteria need to be further described.  1. Please clarify if the study will include or exclude participants with cognitive impairments. 2. For the inclusion criteria for the carer, "Family member or friends regularly engaging with a stroke survivor participant". It's better to state the "Carer" as the primary family caregiver that provides the most or the longest caring job for stroke survivors. 3. Please clarify the sample size calculation strategy in your protocol. 4. For the outcomes and assessment tools, I understand the research group is trying to collect as many outcomes as possible from the participants. However, too many outcomes, especially self-report ones, may take too much time and bring extra pressure on these stroke survivors. I suggest reducing some of the secondary outcomes.
--

VERSION 1 – AUTHOR RESPONSE

Reviewer 1

Dr. M Luz Sánchez-Sánchez, University of Valencia Comments to the Author:

I would like to thank the authors for their interest in addressing a topic of such relevance and at the same time so complex.

I would just like to make some suggestions to improve understanding and the possibility of replicating the study in other countries.

First, the introduction deals with sedentary behavior and physical inactivity; however, the main objective includes extended activities of daily living. I believe that introducing why this variable is included as the main variable would be appropriate. I mean explaining in the introduction why it is relevant in a program that aims to reduce sedentary behaviors.

We thank the reviewer for the above comments and apologise for the lack of clarity around the relevancy / selection of extended activities of daily living as the primary outcome measure for the study. Whilst the immediate intent of our work is to reduce overall time spent sedentary and break up long bouts of sedentary time, as an applied health research project the overall intent is to enhance outcomes for survivors of stroke. Discussions with our patient and public involvement (PPI) group indicated that these are not meaningful outcomes to stroke survivors. However, there is ever

increasing evidence that sedentary behaviour has a detrimental effect on a range of outcomes relating to health and well-being; thus we sought to identify a robust outcome measure that would capture effects of an intervention which targets reducing sedentary behaviour that was meaningful to stroke survivors. We felt ability in extended activities of daily living fulfils this criterion.

After reviewing outcomes in all completed and ongoing trials evaluating sedentary behaviour, further discussions with PPI representatives and members of our programme steering group, we considered the Nottingham Extended Activities of Daily Living ADL scale (NEADL) an appropriate measure. We also think it is important to highlight that reduction of sedentary behaviour is a key secondary outcome in the study (page 11).

We have made added some information to the Introduction section (page 4) to highlight the link between sedentary behaviour and activities of daily living.

Similarly, caregivers are not mentioned in the introduction section either, what would be the hypothesis in this regard? what is the reason for including them as participants in this study? I understand that the caregiver is a fundamental pillar to increase physical activity levels, but why evaluate their burden instead of their involvement/adherence to the program?

We felt it was important to included carers as participants in the study as one of our objectives was to explore whether the intervention reduces carer strain / burden. Our hypothesis was that a reduction in stroke survivors' sedentary behaviour could improve their physical function which in turn could reduce caregiver burden given there is evidence to suggest there is a strong relationship between post-stroke disability and care-giver burden.

We have added some information around this to the Introduction section (page 5).

Secondly, by conducting the study in different centers, have you considered the weight that may have the usual intervention differs greatly from one center to another, have you thought to control for this variable or at least record it?

The reviewer makes an important point. We realise that usual care differs greatly between different stroke services and will be collecting information on the care delivered by both control and intervention sites throughout the trial. We have added some text to reflect this in the 'Intervention' section on page 9. However, as the trial is a pilot study, we will not be incorporating this into the statistical analysis for the participant outcomes.

Thirdly, in relation to the sample size, the reason for not making a calculation should be explained: will the results of this pilot study be used to make a calculation of the subsequent sample size?

We originally calculated the sample size for a definitive phase III trial of 1156 participants. Due to issues related to the worldwide COVID-19 pandemic, recruitment was slower than expected. The revised target sample size for the study of 300-400 participants was based on the estimated recruitment rate during the available time remaining in the funded programme. We have now included this information in the manuscript in a new section titled 'Sample size' on page 20.

Finally, in the manuscript the intervention is explained in a summarized manner, which makes it difficult to replicate the intervention protocol; it would be interesting to have an appendix with more information about the intervention protocol. In this regard, the subsection on the inclusion of a SWAT creates confusion: what does this part have to do with the general intervention? Will all the participants of the intervention group or only some randomly selected ones watch the animation video? If so, it is not explained how this circumstance will be integrated into the statistical analysis.

We appreciate the reviewer's comment that the description of the intervention is insufficient to allow replication; however, we feel it is inappropriate to add further detail at this stage as we hope to progress to a definitive trial and widespread knowledge of the intervention could undermine that. As we state in the intervention section on page 10 of the manuscript, we plan to publish a Template for Intervention Description and Replication (TIDieR) checklist with the trial findings.

With regards the inclusion of the SWAT subsection - the SWAT intervention is designed to increase compliance with the Get Set Go intervention. Clusters allocated to deliver the Get Set Go intervention will be randomly allocated to use the SWAT intervention (Get Set Go plus video animation) or SWAT control (Get Set Go with no video animation). We have amended our description of the SWAT arms in the manuscript to make this clearer (page 10). The SWAT intervention will be delivered at the cluster level. The SWAT will be integrated into the statistical analysis in secondary analyses and will be described in further detail in the statistical analysis plan. Some additional information about how the SWAT will be incorporated into the statistical analysis has been added to the manuscript on page 21.

Reviewer: 2

Dr. Shuanglan Lin, Chongqing Medical University Comments to the Author:

This is a well-designed cluster RCT that aims to evaluate the Get Set Go intervention to reduce sedentary behaviour in stroke survivors. The authors provided a rationale research method. The details of the interventions, application process, outcomes assessments, and data analysis strategies were clarified well in the manuscript. However, some details about the participants' inclusion criteria need to be further described.

1. Please clarify if the study will include or exclude participants with cognitive impairments.

We can confirm that stroke survivors with cognitive impairments may be approached to take part in the study. We have added some additional text within the manuscript in the 'Procedure' section (page 17 and 18) which we hope adds clarity around this point.

2. For the inclusion criteria for the carer, "Family member or friends regularly engaging with a stroke survivor participant". It's better to state the "Carer" as the primary family caregiver that provides the most or the longest caring job for stroke survivors.

We carefully considered the terminology of our inclusion / exclusion criteria for carer participants when drafting our protocol and prior to participant recruitment opening in 2021. Whilst some informal

caregivers may be family members, we also recognise the importance of care provided by those outside of the immediate family, e.g., friends and neighbours; and that care may be provided by multiple people. Our inclusion criteria reflect this.

3. Please clarify the sample size calculation strategy in your protocol.

We originally calculated the sample size for a definitive phase III trial of 1156 participants. However, due to issues related to the worldwide COVID-19 pandemic, recruitment was slower than expected. The revised target sample size for the study of 300-400 participants was based on the estimated recruitment rate during the available time remaining in the funded programme. We have now included this information in the manuscript in a new section titled 'Sample size on page 20.

4. For the outcomes and assessment tools, I understand the research group is trying to collect as many outcomes as possible from the participants. However, too many outcomes, especially self-report ones, may take too much time and bring extra pressure on these stroke survivors. I suggest reducing some of the secondary outcomes.

We thank the reviewer for their comment and acknowledge that they make an important point. We are mindful of the burden on participants and recognise that that the completion of the self-report outcomes may be time-consuming for participants. However, we would like to reassure the reviewer that stroke survivor and carer involvement is central to this research and that we have sought their views on the selection of outcome measures and the questionnaire booklets provided to participants.

The outcome measures have been tested in a feasibility study and participants can be supported to complete them by researchers who provide advice and help where required. It is also worth noting that the questionnaire booklet is divided into smaller sections and participants are encouraged to take breaks between the sections and come back to the booklet at another time if they wish to.

VERSION 2 – REVIEW

REVIEWER	Sánchez-Sánchez, M Luz University of Valencia
REVIEW RETURNED	03-Jul-2023
GENERAL COMMENTS	I would like to thank the authors who have taken into consideration all my suggestions.